# Systematic Review of Human Poisoning and Toxic Exposures in Myanmar

**DOI:** 10.3390/ijerph18073576

**Published:** 2021-03-30

**Authors:** Meghan A. Cook, Pardeep S. Jagpal, Khin Hnin Pwint, Lai Lai San, Saint Saint Kyaw Thein, Thidar Pyone, Win Moh Moh Thit, Sally M. Bradberry, Samuel Collins

**Affiliations:** 1Centre for Radiation, Chemical and Environmental Hazards, Public Health England, Didcot OX11 0RQ, UK; samuel.collins@phe.gov.uk; 2National Poisons Information Service, City Hospital, Birmingham B18 7QH, UK; pardeep.jagpal@nhs.net (P.S.J.); sallybradberry@nhs.net (S.M.B.); 3National Poisons Control Centre, Department of Medical Research, Yangon 11191, Myanmar; khinhninpwint@gmail.com (K.H.P.); lailaisan@gmail.com (L.L.S.); summersaintsaint@gmail.com (S.S.K.T.); 4Global Public Health, Public Health England, London SE1 8UG, UK; Thidar.Pyone@phe.gov.uk; 5Global Public Health, Public Health England, P.O. Box 638, Yangon, Myanmar; Winmohmoh.Thit@phe.gov.uk

**Keywords:** poisoning, toxic, Myanmar

## Abstract

The International Health Regulations (2005) promote national capacity in core institutions so that countries can better detect, respond to and recover from public health emergencies. In accordance with the ‘all hazards’ approach to public health risk, this systematic review examines poisoning and toxic exposures in Myanmar. A systematic literature search was undertaken to find articles pertaining to poisoning in Myanmar published between 1998 and 2020. A number of poisoning risks are identified in this review, including snakebites, heavy metals, drugs of abuse, agrochemicals and traditional medicine. Patterns of poisoning presented in the literature diverge from poisoning priorities reported in other lower-middle income countries in the region. The experience of professionals working in a Yangon-based poison treatment unit also indicate that frequently observed poisoning as a result of pharmaceuticals, methanol, and petroleum products was absent from the literature. Other notable gaps in the available research include assessments of the public health burden of poisoning through self-harm, household exposures to chemicals, paediatric risk and women’s occupational risk of poisoning. There is a limited amount of research available on poisoning outcomes and routes of exposure in Myanmar. Further investigation and research are warranted to provide a more complete assessment of poisoning risk and incidence.

## 1. Introduction

The International Health Regulations (IHR) were adopted by the World Health Assembly (WHA) in 1969 to reduce the negative effects on human health and trade associated with cross-border health threats [1,2]. Under the IHR (1969), countries were required to report to the World Health Organization (WHO) on outbreaks of specific infectious diseases which had the capacity to cross borders, including yellow fever and cholera [1]. In 2005, a revised edition of the IHR was accepted by the WHA, characterised by an all-hazards approach to human health threats [2]. The IHR (2005) required member states to report any threat to human health that had the potential to become a public health emergency of international concern (PHEIC), which could include events that are biological, chemical, or radiological in origin [2].

The revision of the IHR also introduced additional obligations on the part of WHO member states [3]. The IHR (2005) outlined baseline, intersectoral detection and monitoring core competencies that states require to effectively report potential threats to the WHO in a timely manner [4]. These baseline capacities are measured through the Joint External Evaluation (JEE) tool: a voluntary and collaborative assessment of a state’s governance systems and response mechanisms [5]. Using the JEE tool, local authorities work alongside international experts in a given country to assess 49 capacity indicators across 19 technical areas [5]. After considering the available evidence, states are assigned a value between one and five for each indicator, with one indicating the country has ‘no capacity’ in that area, and five indicating ‘sustainable capacity’ [5].

Industrialisation and economic development in low- and middle-income nations has facilitated annual increases in the extraction, production and use of chemicals [6]. Accompanying this growth is an increasing risk of chemical events and emergencies that may cross borders and constitute a PHEIC. However, many developing countries lack established emergency response systems in order to effectively counter these potential risks. As the JEE specifies, one indicator that demonstrates sustainable capacity to detect and respond to chemical events is the presence of a functioning and well-resourced poison centre, or network of poisons centres [5].

Poison centres form part of a state’s public health apparatus, where highly trained staff provide advice in cases of a suspected or actual poisoning following exposure to toxic substance(s). Poison centres may also be attached to laboratories or clinics, which can enable diagnostic and clinical support. If data from enquires made to poison centres are systematically recorded, poison centres can also undertake a surveillance or toxicovigilance function [7]. Data collected by poison centres on the number and nature of poisonings over time can be instrumental in detecting unusual occurrences and novel chemical threats [8,9]. Further, such data provide an evidence base through which to bring about policy changes and regulation to mitigate risks [8,9,10,11,12].

Despite rapid advances in toxicology and its professional representation in Asia since the mid-nineties [13], toxicovigilance activities in Southeast Asia are typically underdeveloped [14]. With limited mechanisms in place to monitor and respond to chemical health threats, Myanmar scored a one out of five for both chemical core capacity indicators during a JEE in 2017 [5,15]. To support international efforts to improve global health security through increased compliance with the IHR (2005), Public Health England, supported by the UK National Poisons Information Service (NPIS), is delivering a programme of work to develop capacities to prevent, detect, and respond to chemical health threats in Myanmar [16]. As part of the ongoing collaboration, Public Health England and the NPIS are working with their counterparts in Myanmar to improve capacity through provision of technical guidance and training of staff at the National Poisons Control Centre (NPCC), three Poison Treatment Units, and affiliated stakeholders. This systematic review was undertaken to identify poisoning priorities identified in the literature and to inform our work in the country. No such review has been conducted in Myanmar. Therefore, this review aims to assess relative risks of toxic exposure, to inform public health initiatives and to identify future research priorities in Myanmar.

## 2. Materials and Methods

Literature searches were conducted to identify causes of poisoning and toxic exposures in Myanmar using the Embase, Global Health, Medline and Scopus databases. A pilot search using the following search string was undertaken in August of 2019:

(((epidemiolog*) OR (incidence) OR (prevalen*) OR (pattern*)) AND ((poison*) OR (sting) OR (bite) OR (snakebite) OR (snake bite) OR ((scorpion) AND (sting)) OR (pesticid*) OR ((poison*) AND (organophosph*)) OR (envenom*). OR (toxic*) OR (intoxic*))) AND (Myanmar OR Burma*)

Following a review of the initial results, a secondary search was conducted to better capture any potential research that explored drug use and heavy metal exposure:

(‘Myanmar’ OR ‘Burma’ or ‘Burmese’) and (‘substance abuse’ OR ‘toxic’ OR ‘venom’ OR ‘sting’ OR ‘toxin’ OR ‘overdose’ OR ‘poison’ OR ‘poisoning’ OR ‘heavy metals’)

Political and economic reforms enacted in Myanmar in the late 20th century meant only articles published from 1998 onwards were considered for inclusion in this study. In 1988, poor economic conditions and low economic development resulted in a series of civil demonstrations and a new military group seizing power. Soviet-style economic planning that had governed national production, private industry and foreign investment was discontinued in favour of market-orientated macroeconomic policies [17]. Between the financial years 1986/87 and 1998/99, participation in the cooperative sector of the economy had declined and the private sector expanded into industries including mining, construction, manufacturing and agriculture. These reforms resulted in 7 percent per annum GDP growth between 1992/3 and 1996/7 [17]. Ongoing participation in trade and economic development meant that Myanmar transitioned from a low-income country to a lower-middle income country in 2015. In accordance with global trends that demonstrate increased chemical production and use in developing countries, increased industrialisation and economic development would have changed the public health risks and poisoning profile of Myanmar [18]. As such, only articles published from 1998 were considered to have been produced in a context comparable to present-day Myanmar, and thereby suitable for inclusion in this review.

The results of both searches were combined, and duplicated results were removed. All results were assessed in the same manner to reduce bias in line with the PRISMA methodology [19], ensuring a transparent and systematic approach to the reviewing process. Articles written in the English language were first screened by title, with remaining papers assessed for their eligibility by abstract and full text (Figure 1).

Full-text copies of all shortlisted papers were available, with major data points and findings extracted from relevant studies. The literature considered for inclusion included studies that examined poisoning or public health effects as a result of exposure to toxic substances in Myanmar. As there are displaced people in border regions of Myanmar, articles that recruited participants from Myanmar in high-risk border settings were also included in this study. Further inclusion and exclusion criteria are detailed in Table 1.

## 3. Results

A total of 2685 unique search results were identified through the literature search, with a total of 34 articles included in the final review (Figure 1). Various causes of human poisoning were identified through the literature, and these are detailed below. An overview of participant demographic data and the context of each study is also provided in Table 2.

### 3.1. Envemonation

Snake bites are a well-documented cause of poisoning in Myanmar, though varied incidence rates have been published by a number of different government departments. A study undertaken by the Department of Medical Research between 1998 and 2000 found yearly prevalence rates of between 17.4 and 24.6 snake bites per 100,000 population [27]. In 2014, the Myanmar Ministry of Health and Sport estimated that the national incidence of snakebite was 29.5 per 100,000 population, with data from health facilities ranging between 19/100,000 and 44/100,000 per region [21]. Another retrospective study examining Department of Health Planning data between 1998 and 2005 estimated the average annual incidence of snakebite in Myanmar to be 8107 bites (R: 6529–9600), with a fatality rate of 7.43% (R:4.93–8.82%) [50]. However, not all of those who experience a snakebite will seek biomedical health treatment, and as such, healthcare data on snakebites are not a true reflection of their actual incidence [21,22]. Accordingly, a 2018 population-based survey in rural Mandalay suggested that the true incidence in that region is likely 116/ 100,000 (95% CI 74/100,000–182/100,000) [21].

Demographic data suggest that men are affected by snakebites at higher rates than women [23,24,25,30], with snakebite cases observed between 50% more [21] to over three and a half times more [28] in men than in women. Different studies suggest the median age of those who experience terrestrial snakebites falls between approximately 27 and 33 years [24,25,27,30]. The literature on sea snake bites in Myanmar also suggests that the majority of those bitten were men, accounting for upwards of 95% of study participants and observed cases [31,33,34]. Across the studies, the weighted average age of a sea snake bite victim was 35.6 years old (R: 16–87 years old) [31,32,33,34].

The context and nature in which people are bitten is highly comparable between studies, with the majority of terrestrial and sea snake bites occurring in occupational settings. Terrestrial snake bites are commonly experienced on the lower limbs [23,24,26,27,30,35] and sometimes the hand or arm [23,24,30,35], often whilst either working on farms [23,26,50] or walking to or from fields [22]. Research from over 4200 households in Mandalay found that in excess of 85% of bites occurred while farming, collecting crops or returning from the fields [21]. Although terrestrial snake bites occur throughout the year, incidence peaks during harvest in the summer months. Incidence spikes again between October and January, during which ploughing activities are undertaken [21,22,24,26,30,35]. As a result, the majority of terrestrial bites occurred during the day [24,25,32,35]. Farmers have indicated that they know boots are an effective preventative measure, but find them costly, too hot to work in, and slow their work by sticking in the mud [22]. Sea snake bites predominantly affect fishermen, generally occurring whilst unloading or sorting fish from netting (though some also occurred while walking or sitting along the seashore) [33]. Likewise, the majority of bites were delivered to the legs, with the next most common site being the hands [33]. However, sea snake bites were more common after sundown, sometimes as a result of poor lighting [33,34].

The vast majority of terrestrial snake bites were attributed to the Russel viper [22,23,25,27,28,35], with bites resulting from cobras [22,23,27,28,35] or green pit vipers [23,27,28,35] also fairly common, alongside a low level of bites from other species [22,23,35]. The acute symptoms that developed following a terrestrial snake bite differed between species, but reported symptoms included pain [23,25], vomiting [23,25], nausea [23], localised swelling [23], acute kidney injury [23,28], necrosis [25], lymphadenopathy [23], ocular manifestations [29] and effects of neurotoxicity [20,21]. Conjunctivitis following the introduction of venom to the eyes by the spitting cobra has also been documented as an outcome [26].

Snake venom toxins are neurotoxic and interfere with blood coagulation, which can result in symptoms ranging from prolonged bleeding to stroke and neurological damage [51]. In the case of Russel viper bites, bleeding is a common and neurological symptoms are much rarer. However, of note are the results of one study amongst agriculture, forestry and fishery workers in the Mandalay region, which found participants were approximately ten times more likely to report neurological symptoms than they were bleeding [20,21]. Studies which examined death as a potential clinical outcome found that between approximately 4.6% and 9.8% of terrestrial snake bites result in fatality [21,23,25,27,35]. Evidence suggests that development of acute kidney injury is positively associated with risk of death [28,30]. Death was also associated with delays in hospital admission, delayed antivenom administration and advanced symptoms [25,27].

Sea snake bites usually resulted in a rapid onset of symptoms, with symptom onset beginning between fifteen minutes and four hours after the bite [33,34]. Commonly reported symptoms included drowsiness [31,33,34], muscle pain [31,33,34], muscle stiffness [31,34], myoglobinuria [31,34], hypertension [34], tachycardia [34], blurred vision [34], and flaccid paralysis [34]. At least one patient who developed myoglobinuria went on to experience renal failure [34]. Of the forty-nine sea snake bite patients detailed in studies where death was recorded as an outcome, six cases were reported to have died as a result of their injury [33,34].

Informal and traditional ethnomedicinal practices, performed by monks or traditional healers, are not uncommon in Myanmar. As such, seeking the attention of faith or traditional healers was reported in studies concerning both terrestrial and sea snake bites [22,26,31,33]. Research indicates that traditional treatment methods are considered effective and are trusted by the community [21,22]. Recommended treatments can include harmless measures such as spiritual chants [22] and consumption of coconut water [21]. However, secondary health risks can result from some practices, including: sucking venom from the wound [21], cutting or tattooing the bite [21], burning the wound [21], rubbing topical applications into the wound [21] and consuming herbal remedies [31,33] (sometimes to induce vomiting [21]). Negative perceptions of the care or treatment provided by biomedical facilities has been documented, though some patients seek the attention of both traditional and biomedical practitioners [22]. Another factor influencing health-seeking behaviour is the relative proximity and low cost associated with traditional healers. Transport to and treatment at biomedical facilities typically requires more time, effort and money on the part of the patient [22]. One study found that people who lived in townships with a hospital that had a good reputation for treating snakebites were less likely to go to a traditional healer [21].

### 3.2. Heavy Metals

Environmental exposure to heavy metals has been attributed to a number of deleterious health conditions including but not limited to kidney disease, cardiovascular disease, neurological impairment and poor development in children [52,53,54]. Studies from Myanmar demonstrate heavy metal exposure and poisoning amongst adults and children, with exposure to lead [37,38], cadmium [38,39], arsenic [36,38,39,40] and selenium [38] documented in populations studied (selenium is not a heavy metal, but a non-metal element that exhibits properties akin to heavy metals). Proposed routes of exposure included occupational settings and environmental contamination [36]. Symptoms in adults have included increased perception of vibrations resulting from lead exposure [37] and altered white blood cell counts following exposure to arsenic [36]. A study on arsenic concentrations in drinking water found adults who had routinely consumed water at or above the WHO reference value of 10 parts per billion [ppb] experienced ‘feelings of weakness’ and ‘chronic numbness or pain’ at higher levels than participants who consuming concentrations of less than 10 ppb (*p* < 0.05) [41]. Adults in this study were also significantly more likely to suffer impaired pain, vibration and two-point discrimination sensory function if they consumed water with an arsenic concentration ≥ 50 ppb [41].

A birth cohort project involving pregnant women in Ayeyarwady showed maternal exposure to heavy metals including arsenic (median: 74 μg/g, IQR: 45–127), cadmium (median: 0.86 μg/g, IQR: 0.50–1.40), selenium (median: 23 μg/g, IQR: 18–30), and lead (median: 1.8 μg/g, IQR: 1.3–3.3) (as measured through creatinine-adjusted urinalysis) resulted in stillbirth, premature birth, low birth weight and congenital abnormalities [38]. Pre-natal exposure to arsenic and cadmium also resulted in significantly shorter leucocyte telomere length in the newborns [39]. Maternal urinary arsenic concentrations correlated with arsenic concentrations in household drinking water (*p* < 0.001) [40]. Further, higher urinary arsenic concentrations were significantly associated with urinary levels of 8-hydroxydeoxyguanosine, a biomarker of arsenic-induced oxidative stress [40].

A study examining blood lead concentrations among displaced children in camps on the Thai-Myanmar border found that 5.1% of participants had elevated blood concentrations (≥10 µg/dL), rising to 14.5% of children under 2 years of age [42]. Some risk factors associated with elevated blood lead concentrations in children under 2 years of age included exposure to car batteries used for power generation in the camps, cosmetic products and traditional medicine such as ‘Tum Shwe War’ chest rub, especially if children had mouthed the items [42]. Several traditional medicines and cosmetics available in the camp were tested for heavy metals, with dangerously high levels found in a multipurpose infant remedy ‘Gaw Mo Dah’ (525 ppm lead, 20,500 ppm arsenic) and a remedy called ‘Wonotsay’ (26,200 ppm arsenic) [42].

In one study involving adult men in Ayeyawady exposed to arsenic through contaminated groundwater, 59% of participants had blood arsenic concentrations of >0.025 µg/mL [36]. Another study involving adult men found workers from small-scale battery workplaces in Yangon had blood lead concentrations significantly higher (4.25 ± 3.87 µg/dL, *p* = 0.007) than the non-exposed control group (2.14 ± 1.02 µg/dL) [37].

One study also examined adult literacy across Myanmar and found lower levels of literacy in townships with lead mining activities, even amongst those with access to high-quality sanitation infrastructure, which may indicate wider environmental exposure to mining by-products [55].

### 3.3. Alcohol and Drugs of Abuse

Myanmar is a prominent producer of narcotic drugs in Southeast Asia, with north-east regions of the country that share a border with China, Lao People’s Democratic Republic and Thailand forming part of the ‘Golden Triangle’. Shan State in particular, located in Myanmar’s north-east border region, has been a major site of heroin and opium production [43,56,57]. However, in recent decades, Myanmar has also become an increasingly prominent producer of synthetic drugs, and is now one of the biggest producers of methamphetamine pills in the region [56,58]. Ketamine is also being seized at increasingly higher rates, with national drug enforcement data indicating that the volume of ketamine seized by authorities increased by more than 560% between 2014 and 2018 (4.2 kg to 2360.2 kg) [59].

While drugs produced in Myanmar are primarily sold in international markets, some domestic use does occur [60]. The prevalence of domestic drug use is suspected to be higher in border regions, as residents typically have higher incomes from migrant work, drug availability is higher and prices are lower [60]. In 2013, the Myanmar Central Committee for Drug Abuse Control (CCDAC) estimated that there were between 300,000 and 400,000 drug users in the country [56]. However, no comprehensive national surveys on drug use amongst adults or adolescents have ever been undertaken by the Myanmar government [60]. Furthermore, drug use patterns may not reflect drug dependence, with a 2005 report published by the United Nations Office on Drugs and Crime (UNODC) indicating that in many border villages of Shan State, opiates were taken for medicinal reasons in the absence of formal medical care [57].

Reports published by the UNODC since 2005 suggest that, overall, heroin and opium use in Myanmar has declined or remained relatively stable, perhaps due to increased drug enforcement in 1999 and subsequent decreases in opium production [57,58]. However, the use of heroin and opium did appear to increase again in 2017 and 2018 [59,60], and most people admitted to drug treatment centres in Myanmar are opiate users. Of those admitted to drug treatment centres, the majority are aged between 20 and 39 years old (76%) [59]. UNODC reports suggest that methamphetamine use in Myanmar has increased each year since 2005 [56,60]. While methamphetamine production in Myanmar is also largely undertaken in Shan State [43], drug enforcement statistics from 2009 demonstrate that methamphetamine pills are available throughout the country, with seizures reported in sixteen of Myanmar’s seventeen administrative regions [60]. Since 2013, an increasing proportion of users admitted to drug treatment centres have identified as users of methamphetamine or ‘other drugs’ [59].

Two systematic reviews that estimated global levels of injected drug use, published in 2004 and 2008, suggested that Myanmar could have somewhere between 60,000 and 300,000 injected drug users [60,61,62]. However, neither systematic review cited publicly available source data that informed these estimates for Myanmar. A 2014 study that recruited Myanmar men over 15 years of age estimated 83,314 (95%, 55948-113021) men inject drugs nationally [46]. This study also examined injecting drug use in women, but the researchers suggested that recruiting women into the study was more difficult and they lacked sufficient data to estimate associated prevalence rates [46]. Northern regions including Kachin state, Shan state and the Sagaing region were estimated to have the highest prevalence of people who inject drugs, at approximately 4.12% of the population [46]. A 2013 UNODC report suggests that less than 1% of methamphetamine users in Myanmar administer the drug by injection, which would indicate that the vast majority of those injecting drug are using opiates [56].

In 2005, a country profile published by the UNODC suggested that higher methamphetamine use has been observed in particular occupations such as truck drivers and sex workers [57], and that young drug users are more likely to use amphetamines [56,57]. One 2017 study in Muse, a town in northern Shan state on the border with China, studied methamphetamine users (aged 18 to 35 years old) in various occupations including students, labourers, housewives, highway drivers and female sex workers (FSWs). This study found the median age of first use was 16 years (R: 10–29 years) for males and 17 years (R: 9–27 years) for females [43]. Early onset of methamphetamine use was reported at higher rates amongst bisexual or homosexual men and amongst those whose first use was at an entertainment venue [43]. Reasons for first methamphetamine use were ‘encouraged by friend/sexual partner/drug dealers’ (41.23%), ‘for weight loss/work-related purposes’ (32.37%), and ‘curious about methamphetamine effects/for fun’ (26.40%) [43]. Another study from Muse that recruited 101 Myanmar FSWs found that approximately one third of participants (34%) indicated current drug use, with only one user reporting drug injection [47]. FSWs engaging in current drug use were slightly older than non-drug users (*p* = 0.015), had been involved in sex work slightly longer (*p* = 0.005), were more likely to have previously worked in other entertainment venues (*p* = 0.026), to have a previous partner who used drugs (*p* = 0.003), and to have crossed the border into China (*p* = 0.018) [47].

Quality data on levels of alcohol use or risk of alcohol poisoning are also lacking. Research on alcohol use amongst 13- to 15-year-old adolescents across multiple countries found that Myanmar adolescents (50.8% male) were the least likely to self-report consumption of alcohol in the previous 30 days (1.6%), or to have ever been drunk (3.0%) [44]. To assess levels of alcohol use amongst Myanmar residents of a Thailand-based camp, one study recruited displaced, pregnant Myanmar women to find the frequency of high-risk drinking behaviour (defined as six or more standard drinks in a session) [45]. This study used a Short Assessment Screening Questionnaire (mSASQ) in order for participants to self-report their own behaviours (in the previous year prior to falling pregnant) and the behaviour of their male partners (for the duration of the previous year). High-risk drinking behaviour as self-reported by female participants was extremely low (0.2%), though participants indicated high-risk alcohol use was more common amongst male partners (24.4%). This study suggested that alcohol consumption amongst women and consumption to the point of intoxication in men was stigmatised in this setting, which may lead to underreporting [45]. However, other research indicated that alcohol and drug abuse in camps on the Thai–Myanmar border is commonplace [63], with use of alcohol a culturally acceptable response to the stress caused by displacement, particularly for men [64].

Traditional plants such as Kratom, a plant that interacts with opioid receptors in the brain and has stimulatory effects, are also used for their psychoactive properties in Myanmar, but their study is limited [56].

### 3.4. Agrochemicals

Exposure to pesticides can lead to acute or chronic poisoning, with outcomes ranging from irritation of the skin and the eyes to impaired development in children, reproductive disorders, cancer and death [65]. In one study of male groundnut farmers in Mandalay, 400 participants were recruited as part of an observational study on pesticide use and exposure. The study suggested that the use of organophosphate (OP), organochlorine, and carbamate pesticides were common in Myanmar, but that most groundnut farmers used OP pesticides [48]. Furthermore, the study found that many mixed more than one OP pesticide for use on their crops, exceeded the recommended dose, did not adequately dispose of empty containers, and would take breaks in their working environment. Personal protective equipment (PPE) was not used by any of the farmers, who indicated that it was not suited to a tropical environment, it was uncomfortable to wear for extended periods, that it was expensive, and it was not widely available.

From the 400 participants in the observational study, 100 men (37.5 years ± 9.45 (R: 18–49)) were selected to provide blood and semen samples for analysis [48]. Analysis of blood and semen samples was used to track how increased use of pesticides during the growing season affected health outcomes as compared to the non-growing season when pesticide use is limited. The proportion of participants who had a low sperm count rose from 46% in the non-growing season to 74% during growing periods (*p* < 0.05). Other biomarkers of seminal quality that worsened between the non-growing and growing season (*p* < 0.05) were viscosity, morphology and sperm motility. Analysis of the blood samples also demonstrated declines in serum hormone concentrations and an increase in blood-cholinesterase levels between the non-growing and growing season [48].

### 3.5. Traditional Medicine

In addition to the heavy metal contamination of some traditional medicines already detailed, traditional medicines can present their own health risks for poisoning. In one case study, a 34-year-old man developed acute liver injury after consuming 700 mL of a remedy called ‘Dan Ywet’ for three consecutive days. A common remedy made of boiled henna leaves, ‘Dan Ywet’ is purported to have astringent, antibacterial and anti-infective properties. The man presented to hospital two weeks after consuming the Dan Ywet with symptoms including mild jaundice, dark-coloured urine, shortness of breath on exertion and dizziness. On clinical examination, he was febrile (37.8 °C), tachycardic (100 bpm) and hypertensive (150/90 mmHg). He became anuric after two days in hospital. The patient was treated with fluid replacement, antibiotics, antiemetics, and H2-receptor blockers. On day four, the patient commenced haemodialysis treatment (creatinine 1418 μmol/L). The patient made a complete recovery and was discharged from hospital on day 38. Serum creatinine on discharge was 245 μmol/L [49].

## 4. Discussion

This review is the first systematic assessment of the literature pertaining to human poisoning in Myanmar. Both acute and chronic causes of poisoning were identified. The most documented cause of poisoning was envenomation, with approximately half of the peer-reviewed literature suitable for inclusion pertaining to snakebite prevalence and management. Other causes of poisoning identified in Myanmar included heavy metals, drug abuse, agrochemicals and traditional medicine. However, the number of studies in these areas was limited and there were few clinical studies. A significant portion of the available peer-reviewed research studied high-risk populations in high-risk settings, including pregnant women, sex workers, drug users and displaced people in border regions or camps. Research on the public health risks experienced by high-risk and marginalised populations in other areas of Myanmar, or on the public health risk of occupational and household poisoning in Myanmar more broadly, was lacking. Of note was a complete absence of studies on acute exposure to pesticides or toxic alcohols.

The pattern of poisoning in Myanmar presented by the literature in this review diverges from publications examining other developing nations in Southeast Asia, including a retrospective study conducted in Vietnam (1998–2003) and a 2013 literature review from Thailand. In Vietnam, the leading causes of poisoning were identified as ‘food/beverages’ (35%), ‘pharmaceuticals’ (33%), ‘snakes/other poisonous animals’ (12.6%), ‘pesticides/rodenticides’ (9.2%) and ‘drugs of abuse’ (4.6%). In Thailand, 2006–2008 data indicated that poisonings primarily involved ‘pharmaceuticals’ (38.8%), ‘bites and stings’ (31.7%), ‘household products’ (17.6%) and ‘insecticides’ (3.3%). In both Vietnam and Thailand, increasing urbanisation resulted in decreased agricultural chemical exposures over time. In Vietnam, service workers and students became the groups most as risk of poisoning, with the highest number of poison centre patients aged between 15 and 24 years old. In Thailand, those under 30 years of age were also disproportionately represented in poisoning cases. Both countries also found that women were highly represented in the poisoning data, with 50% of poisoned patients in Vietnam being women, and the Thai review concluding that women faced an equal or greater risk of poisoning than men. The vast majority of poisonings in Vietnam occurred in the home (74.1%), and in Thailand exposures in the home were trending upwards. In both countries, self-harm and attempted suicide were a leading cause of poisoning, attributed to one-third of poisonings in Vietnam and 50–66% of poisonings in Thailand.

There were large gaps in the Myanmar poisoning literature, particularly with reference to some of the higher risks identified in Vietnam and Thailand. In nine of the thirty papers and case studies in which demographics of Myanmar participants were provided, upwards of 95% of the participants and patients reported on were male. Available research and case studies with near exclusive male representation examined a relatively broad scope of poisoning including snakebites, drug injection, occupational risks and arsenic exposure in a variety of contexts. In contrast, five papers recruited only female participants: three were associated with one birth-cohort project examining arsenic exposure in pregnant women, one that recruited displaced pregnant women to assess alcohol use among participants and their male partners, and one study on drug use in female sex workers. Research on poisoning risks amongst children and adolescents was also low, with only one study that focused on poisoning in Myanmar children exclusively: a study on lead poisoning in a Thai-Myanmar camp. There were no publications on the public health impact of pharmaceuticals or poisoning with household chemicals. Furthermore, no paper examined the public health burden of self-harm and suicide, perhaps as a result of suicide being a crime in Myanmar.

Low levels of systematic toxicovigilance and chemical event monitoring in Myanmar likely means that cases of poisoning are underreported. A set of recently drafted National Poison Management Guidelines [66] lists the following major causes of poisoning in Myanmar, as determined by admissions to the New Yangon General Hospital Poison Treatment Unit (NYGH PTU): organophosphorus pesticides, paraquat, chlorpheniramine, paracetamol, methanol, narcotics, benzodiazepines, corrosives, rodenticides, and petroleum products. Envenomation is not included as snakebite patients are typically not treated at the NYGH PTU. Though not peer-reviewed, the findings of a 2017 Master’s thesis that focused on poisoning outcomes in NYGH PTU support the poisoning risks as they are presented in the National Poisons Management Guidelines. The thesis used interviews and medical records to conclude that those aged 16 to 25 years were at the highest risk of poisoning (42.7% of patients), that most poisonings occurred in the home (74%) and that gendered risk was roughly equal (53.1% female). Furthermore, the most common type of poisoning was as a result of drug overdose (46.9%) and nearly three-quarters of poisonings (74%) were intentional (poisonings under the category ‘drug overdose’ were primarily pharmaceutical overdoses (86.7%), though it also included drugs of abuse (8.9%) and traditional medicine (4.4%)) [67]. The available literature on poisoning in Myanmar does not reflect these professional insights. These findings align with poisoning priorities published in Vietnam and Thailand, highlight significant gaps in peer-reviewed research, and suggest several priority areas for further research to inform a deeper understanding of the true patterns and prevalence of poisoning in Myanmar.

Although the data presented in this review are limited, it can be used to facilitate capacity-building activities and training within the poisoning management system in Myanmar. Firstly, the lack of data should be used to raise awareness that a systematic method of collecting information about the prevalence and causes of poisoning is required. The development of a robust national toxicovigilance system to monitor poisoning data can play a pivotal role in facilitating prevention activities. For example, it has previously been demonstrated that acute poisoning can be prevented (or the impacts reduced) by systematic analysis of the contexts, causes and populations at risk of poisoning [68]. Furthermore, toxicovigilance can facilitate the detection of novel and emerging threats through the development of an efficient alerting system. Geographical analyses of the types of poisoning occurring in different areas, as well as how poisoning differs between rural and urban contexts, can help to inform prevention and response at the level of townships and station hospitals. All of these areas are critical elements of IHR compliance.

A number of health risks have been highlighted that would warrant more detailed controlled studies, including the poisoning risk associated with pesticides and pharmaceuticals. The National Poisons Control Centre, in collaboration with the three Poison Treatment Units in Myanmar, would be well-placed to conduct studies to address these needs. Further researching and understanding these needs would help the NPCC develop appropriate resources for the identification and treatment of poisoning, such as tailored poisoning monographs. In addition, there were few data relating to the types of products causing poisoning. Identification of common products of concern can facilitate better regulatory measures and control of the risks.

## 5. Conclusions

This review on poisoning in Myanmar has highlighted a number of poisoning risks among the national population, including snake bites, heavy metals, alcohol, drugs of abuse, agrochemicals and traditional medicine. However, perhaps more importantly, it highlighted significant gaps in the available peer-reviewed research on areas including but not limited to pharmaceuticals, occupational exposures, household exposures and self-harm. Data on poisoning in Vietnam and Thailand correlate with the poisoning priorities highlighted by professionals who work at the poison treatment units in Yangon. Their data and experience suggests that young people are the most likely to experience poisoning outcomes, often as a result of self-harm using pharmaceuticals or as a result of household exposures. However, more high-quality research on poisoning in Myanmar will need to be undertaken to better understand actual poisoning patterns. This research is crucial to building capacity in the healthcare sector to better detect, survey and respond to these domestic chemical health threats. Furthermore, these capacities underscore compliance with the IHR to prevent public health emergencies of international concern.

## Figures and Tables

**Figure 1 ijerph-18-03576-f001:**
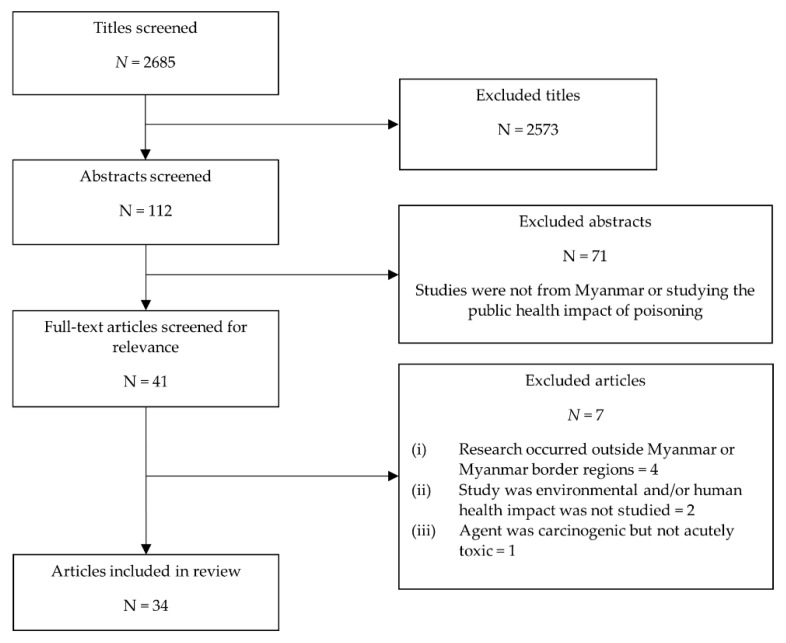
Flow chart showing inclusion and exclusion of articles.

**Table 1 ijerph-18-03576-t001:** Inclusion and exclusion strategy.

Criteria Topics	Include	Exclude	Rationale/Comment
Literature	Peer-reviewed literature	Literature that has not been peer-reviewed	Peer-reviewed literature is the academic standard
	Grey literature of high quality	Low-quality grey literature	High quality grey literature is a useful and trusted source of information.
	Epidemiological studies and case studies	Studies on community perception of risk	Epidemiological studies and case studies on poison outcomes demonstrate that poisoning is experienced in Myanmar. Papers that focus on community perceptions of risk or social constructs around poisoning only were excluded from the review.
	English language literature	Literature in a language other than English	English is a common second language in Myanmar and university courses are often taught in English. As such, articles published in the Myanmar Health Sciences Research Journal are written in English, and the exclusion of articles written in a Myanmar language did not significantly impact the availability of locally produced research.
Poisoning	Acute and chronic poisoning as a result of short- or long-term exposure to a toxic substance.	Acute and chronic exposure to the majority of biological agents.	Toxic substances will include exposure to pharmaceuticals, drugs of abuse chemicals and envenomation. Reports of adverse reactions to therapeutic does of drugs (rather than overdose) excluded.
	Food and water exposures		Exposure to heavy metals such as lead and mercury through food or water sources was included in the review. Food poisoning as a result of toxins produced by foodborne bacteria excluded from the study. Food poisoning only included as a result of chemical contamination of food.
Myanmar	Articles written from border regions that have been displaced from Myanmar.	Myanmar people living in other countries outside of border regions.	Articles on Myanmar people who have migrated to other countries, or on poisoning as a result of Myanmar animal species kept as pets in other countries, have been excluded.

**Table 2 ijerph-18-03576-t002:** Study location and participant demographic data.

Study	Poisoning	Region	Context or Population	Number of Participants	Majority Participant Gender	Age	Self-Harm Studied?
Mahmood et al. 2019 [20,21]	Snakebite	Mandalay	Farming townships	4276	50.1% Male	Primarily 18+	N/A
Schioldann et al. 2018 [22]	Snakebite	Mandalay	Farming townships	135 (7 *)	57.14% Male *	12 to 45 years *	N/A
Mahmood et al. 2018 [21]	Snakebite	Mandalay	Farming townships	4276	50.1% Male	Primarily 18+	N/A
White et al. 2019 [23]	Snakebite	Mandalay	Mandalay General Hospital	948	61.2% Male	18+	N/A
Myo-Khin et al. 2012 [24]	Snakebite	Mandalay	Nahtoogyi Township Hospital	101	69% Male	Mean 32.2 (SD: 15.5, range: 3–80 years)	N/A
Pe et al. 2002 [25]	Snakebite	National	Six township hospitals from five snakebite endemic divisions	294	77% Male	Average 27 years (range: 7–75 years)	N/A
Pe et al. 2005 [26]	Snakebite	Magway and Mandalay Region	Taungdwingyi and Kyaukpadaung townships	1381 *	64.8% Male *	Mean: 30 years (range:3–84 years) *	N/A
Aye et al. 2018 [27]	Snakebite	Yangon	Tertiary hospitals	246	80.0% Male	Median 31 (IQR: 23–42)	N/A
Aye et al. 2017 [28]	Snakebite	Yangon	Tertiary hospitals	258	78.7% Male	Median 31 (IQR: 23–42)	N/A
Aye, Naing and Myint, 2018 [29]	Snakebite	Magway	Case study	1	Male	70 years	N/A
Thien and Byard, 2019 [30]	Snakebite	Magway	Magway Region General Hospital	84 ^†^	64% Male ^†^	Age range 5–75 years, mean 33 years ^†^	N/A
Pe et al. 2005 [31]	Sea snake bite	Mon	Fishing communities	46	>97% Male	12–65 years	N/A
Pe et al. 2006 [32]	Sea snake bite	Yangon, Mon and Ayeyawady	Fishing communities	187 *	89.3% Male *	Mean 35.64 years (range: 10–87 years) *	N/A
Mya et al. 2005 [33]	Sea snake bite	Yangon	Fishing communities	47	95.7% Male	16–87 years	N/A
Myint, Pe and Mya, 2006 [34]	Sea snake bite	Yangon	Fishermen	2	Males	21 and 56 years old	N/A
Myint, Pe and Maw, 2002 [35]	Snakebite	National	Healthcare data	Not provided	Not provided	Range: 6–77 years	N/A
Thu et al. 2010 [36]	Arsenic	Ayeyawady	Community members	70	100% Male	18–50 years old	N/A
Oo et al. 2018 [37]	Lead	Yangon	Men working in small-scale battery workplaces	56	100% Male	24–45 years	N/A
Wai et al. 2017 [38]	Heavy metals	Ayeyawady	Pregnant women and newborns	419	100% Female	Mean: 28 years old (SD: 6.6 years)	N/A
Wai et al. 2018 [39]	Heavy metals	Ayeyawady	Pregnant women and newborns	409	100% Female	Mean: 28 years old (SD: 6.6 years)	N/A
Wai et al. 2019 [40]	Arsenic	Ayeyawady	Pregnant women	198	100% Female	Mean: 28 years old (SD: 6.6 years)	N/A
Mochizuki et al. 2019 [41]	Arsenic	Ayeyawady	Thabaung Township	1867	61% Female	Mean: 35.2 ± 20.4	N/A
Mitchell et al. 2012 [42]	Lead	Thailand-Myanmar border	Displaced children	642	51.7% Male	Range: 6 months to 14 years	
Saw et al. 2017 [43]	Methamphetamine	Shan	Students, laborers, housewives, highway drivers, female sex workers (FSW) and men who have sex with men (MSM) near the Chinese border	1362	56.9% Male	18–35 years	No
Balogun et al. 2013 [44]	Alcohol	Not provided	Students	2804	50.8% Male	13–15 years	No
Ezard et al. 2012 [45]	Alcohol	Thailand-Myanmar border	Displaced pregnant women	636	100% Female	Median 26.2 years (range: 15 47 years)	No
Johnston et al. 2018 [46]	Drugs of abuse	National	Injecting drug users	~3920 §	>99% Male	≥15 years	No
Hail-Jares et al. 2016 [47]	Drugs of abuse	Shan	FSW	101	100% Female	Median 25 years (IQR 22–28)	No
Lwin et al. 2018 [48]	Organophosphate Pesticides	Mandalay	Groundnut farmers	400 (100 ^‡^)	100% Male ^‡^	Mean 37.5 years ±9.45 years (range: 18–49 years) ^‡^	No
Khine, 2017 [49]	Traditional medicine (Dan Ywet)	Yangon	Hospital patient	1	Male	34 years old	No

* Demographic data for snake bite victims only, excluding focus group or wider community participants. ^†^ Post-mortem study. § Estimate based on sample size required from each township, final number of participants not provided. ^‡^ Participants selected to provide blood samples for laboratory analysis.

## Data Availability

Data sharing is not applicable to this article.

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
