# Peer review of "Systematic Review of Human Poisoning and Toxic Exposures in Myanmar"

_ijerph, 2021, doi:10.3390/ijerph18073576_

Round 1

Reviewer 1 Report

The manuscript entitled „Systematic review of human poisoning and toxic exposures in Myanmar” presents a literature search which was undertaken to find articles pertaining to poisoning in Myanmar published between 1998 and 2020. Based on the literature findings the authors identified poisoning risk, including snakebites, heavy metals, drugs of abuse, agrochemicals and traditional medicine. 
The manuscript is an interesting contribution as it especially highlights the gaps related to the poisoning and routes of exposure. 
The authors focused on a specific study site Myanmar, but I cannot see any details about the study area. I expect to see the description of the study site and the justification why this study site is important when it comes to the poisoning analysis. 
I also suggest to improve the quality of the results presentation. There is a lot of numbers within the text, but in my opinion it would be valuable to prepare some graphs presenting the most important findings. Some interesting graphics would atract the readers attention, because currently it is hard to follow the intention of the manucript with a so significant load of information given. 
The manuscript has a potential for readers but before publishing it must be amended thorougly.

Author Response

Reviewer one:

Thank you for your kind comments.

We have chosen to conduct a literature review of all academic research conducted in the country of Myanmar. No review of this health priority has been undertaken in Myanmar, and surveillance of poisoning issues in Myanmar is underdeveloped. We have undertaken this review in order to inform our work in the country, which centres around strengthening poison-related health services. We would like to clarify that this review does not contain any original research conducted by the authors at a particular field site.

We appreciate that the review contains many references in numbered format alongside data from research papers. To try and provide clarity for readers, we’ve presented an overview of the research articles that constitute our review in a table which indicates the type of poisoning, study sites and demographics involved in the research. This table is the first figure in the results section. As the research presented in this review spans multiple areas of poisoning and different research methodologies, providing a visualisation of the results in a graph format would be quite difficult.

Reviewer 2 Report

In this study, the authors have assessed poisoning and toxic exposures in Myanmar between 1998 and 202 by undertaking the systematic literature search. Their study emphasized significant gaps in the available peer-reviews research on areas including but not limited to pharmaceuticals occupational exposures, household exposures and self-harm. This work is significantly essential to assess relative risks of toxic exposure, and it can provide useful information for the public health initiatives and future research priorities in Myanmar.

Generally, this study presents some important data on building capacity in the healthcare sector of Myanmar. However, before that, the following major points should be carefully responded and revised.

Specific comments:

  1. In line 26, “Further investigation and research is” should be corrected as “Further investigation and research are”.
  2. In line 63, “If data from enquires made to poison centres are systematically” should be corrected as “If data from enquires made to poison centres is systematically”.
  3. In line 69-70, “Despite rapid advances in toxicology and its professional representation in Asia since 69 the mid-nineties” should be corrected as “Despite of rapid advances in toxicology and its professional representation in Asia since 69 the mid-nineties”.
  4. In line 108, “increased industrialisation and economic development” should be corrected as “increasing industrialisation and economic development”.
  5. In figure 1, please cut off the line at the top of the figure.
  6. In table 1, “Food and water exposures” of the first column should be centered.
  7. Table 2 lacks annotations.
  8. If not necessary, reference should be placed at the end of the sentence.
  9. In line 189-190, “Studies in which death from a terrestrial snake was recorded” should be corrected as “Studies in which death from a terrestrial snake were recorded”.
  10. In line 233-234, “74ug/g” “23ug/g” should be corrected as “74 ug/g” “23 ug/g”. The same problems below should be corrected.
  11. In line 255, “2.14± 1.02 μg/dl” should corrected as “2.14 ± 1.02 μg/dl”. The same problems below should be corrected.
  12. In line 313, “35-years-old” should be corrected as “35-year-old”.
  13. In line 329, “13 to 15-year-olds” should be corrected as “13 to 15-year-old”.
  14. In line 377, “(37.8C)” should be corrected as “(37.8 oC).

Author Response

Reviewer two:

Thank you for your kind comments. We appreciate the thorough feedback on the manuscript, and we’ve provided a line by line response below:

  1. In line 26, “Further investigation and research is” should be corrected as “Further investigation and research are”.

Thank you, this has been corrected

  1. In line 63, “If data from enquires made to poison centres are systematically” should be corrected as “If data from enquires made to poison centres is systematically”.

Thank you, this has been corrected

  1. In line 69-70, “Despite rapid advances in toxicology and its professional representation in Asia since 69 the mid-nineties” should be corrected as “Despite of rapid advances in toxicology and its professional representation in Asia since 69 the mid-nineties”.

Thank you for your suggestion but we feel the original sentence is grammatically sound, unless we were to change the start of the sentence to ‘In spite of’.

  1. In line 108, “increased industrialisation and economic development” should be corrected as “increasing industrialisation and economic development”.

Thank you for your suggestion but we’d like to keep ‘increased’ to show past-tense and demonstrate the change in poisoning patterns has already occurred. However, we note that you are right to try and demonstrate that Myanmar has continued to develop and industrialise throughout the study period.

  1. In figure 1, please cut off the line at the top of the figure.

Thank you, this has been corrected

  1. In table 1, “Food and water exposures” of the first column should be centered.

Thank you for your suggestion. We’d like to clarify that ‘food and water exposures’ span both the ‘include’ and ‘exclude’ column based on the content of the research. As such, ‘food and water exposures’ is centred, though the differences in width between the ‘include’ column and the ‘exclude’ column make it appear slightly out of line.

  1. Table 2 lacks annotations.

Thank you, this appears to have been lost in updates to the formatting of the paper. We have attempted to add the title of the table back in, but cannot do so without altering formatting revised by journal. We’ve added in the following title at the bottom of the table and would ask that the journal please amend as necessary: ‘Table 2: Study location and participant demographic data’.

  1. If not necessary, reference should be placed at the end of the sentence.

Thank you for the suggestion. However, some sentences cover a variety of information not covered in every research article referenced in that sentence. For instance, we may have a number of articles covering symptoms of snake bites, but only some of those articles mention vomiting as a symptom. To make this clear to the reader, and to make it easier for other researchers to identify research articles that are relevant to their study area, we have tried to place different citations as accurately as possible. This convention is not uncommon in review articles and we hope that it helps indicate the relative weight of evidence. For instance, if there are 6 articles that reference vomiting and only one article that references neurological symptoms, the reader can easily deduce that vomiting is a more frequently observed symptom.

  1. In line 189-190, “Studies in which death from a terrestrial snake was recorded” should be corrected as “Studies in which death from a terrestrial snake were recorded”.

Thank you for your comment. We’ve revised the sentence to read: ‘Studies which examined death as a potential clinical outcome found that between approximately 4.6% and 9.8% of terrestrial snake bites result in fatality’.

  1. In line 233-234, “74ug/g” “23ug/g” should be corrected as “74 ug/g” “23 ug/g”. The same problems below should be corrected.

Thank you, this has been corrected

  1. In line 255, “2.14± 1.02 μg/dl” should corrected as “2.14 ± 1.02 μg/dl”. The same problems below should be corrected.

Thank you, this has been corrected

  1. In line 313, “35-years-old” should be corrected as “35-year-old”.

Thank you for your comment. We’ve revised the sentence to read: ‘…studied methamphetamine users (aged 18 to 35 years old)’

  1. In line 329, “13 to 15-year-olds” should be corrected as “13 to 15-year-old”.

Thank you for your comment. We’ve revised the sentence to read: ‘Research on alcohol use amongst 13 to 15-year-old adolescents across multiple countries..’

  1. In line 377, “(37.8C)” should be corrected as “(37.8 oC).

Thank you, this has been corrected

Reviewer 3 Report

This manuscript is a Review  and has 9 authors.  They have assembled a coherent, well-written interesting  manuscript on a topic of poisoning in Myanmar that deserves publication

The comments below are mainly merely like thoughts.

Page numbering not ordered, but there are 16 pages.

Line 52  Interestingly, I guess the verb could be “has” or “have”.

Line 72 “both chemical capacity indicators” -  Not clear to me if it means monitor and respond?  Respond is not an indicator.  What are the two indicators?

Lines 97 - 112 only articles later that 1998 used in the study.

Line   Page 1 of 16 is really page 5.  The table, Table 2, has no Title.

Line 136  Envemonation,  had to look it up.

Lines 137-218    Data on snake bites very interesting.

Lines 202-17 treatment snake bites

Lines 219 - 259 heavy metals.  Like comment on Se being a non-metal (below Line 238), it is a metalloid.

260-347 Drinking and drugs  Excellent section which most all countries have data on.

Line 328   however “quality data on levels of alcohol use or risk of alcohol poisoning are also lacking.”  Any Ideas?

Lines 348-368  Agrochemicals  Short section, but has one emphasis on organo phosphate fertilizers yielding low sperm count.

Lines 369 -  382  Traditional Medicine  short  One case study of Dan Ywet.

Lines 383 – 474 Discussion  Identifies that the manuscript is the “first systematic assessment of the literature pertaining to poisoning in Myanmar.”  Included is data on health issues from Vietnam, Thailand.  This section discusses the results of a recent thesis which focuses on the reports from the New Yangon General Hospital, Poison Treatment Unit’s records.  The Unit does not generally treat snake bites, so data missing on this topic.  Most common treatment is for drug overdose.

Line 446 (there is s 2 in the word overdose2).

L   475-489  Conclusions  Looks to me that the conclusions point o the is a need for more quality research with peer reviewed reporting.

Author Response

Reviewer three:

Thank you for your kind comments. We appreciate the thorough feedback on the manuscript, and we’ve provided a line by line response below:

Page numbering not ordered, but there are 16 pages.

Thank you, we have noticed that the page numbering has been affected by the section breaks as a result of formatting requirements. We hope that the journal will be able to correct this when they undertake the final copy-editing process.

Line 52 Interestingly, I guess the verb could be “has” or “have”.

Thank you for your feedback. We’ve amended the sentence slightly to: ‘Industrialisation and economic development in low- and middle-income nations has facilitated annual increases in the extraction, production and use of chemicals’. As they’re separate but inter-related trends, we hope you agree that the use of ‘has’ is still agreeable.

Line 72 “both chemical capacity indicators” - Not clear to me if it means monitor and respond?  Respond is not an indicator.  What are the two indicators?

Thank you for your feedback. We’ve tried to introduce concepts central to the JEE tool in the second paragraph of the Introduction, which is split over page 1 and page 2. It reads:

These baseline capacities are measured through the Joint External Evaluation (JEE) tool: a voluntary and collaborative assessment of a state’s governance systems and re-sponse mechanisms [5]. Using the JEE tool, local authorities work alongside interna-tional experts in a given country to assess 49 capacity indicators across 19 technical areas [5]. After considering available evidence, states are assigned a value between one and five for each indicator, with a one indicating the country has ‘no capacity’ in that area, and five indicating ‘sustainable capacity’ [5].

‘Chemical events’ is one of the 19 technical areas and the two associated indicators cover detection/monitoring and response priorities. The two capacity indicators are as follows:

CE.1 Mechanisms are established and functioning for detecting and responding to chemical events or emergencies.

CE.2 Enabling environment is in place for management of chemical Events

Lines 97 - 112 only articles later that 1998 used in the study.

Thank you for your feedback. We’ve amended the sentence to say: ‘Political and economic reforms enacted in Myanmar in the late 20th Century meant only articles published from 1998 onwards were considered for inclusion in this study.’

Line   Page 1 of 16 is really page 5.  The table, Table 2, has no Title.

Thank you, we have noticed that the page numbering has been affected by the section breaks as a result of formatting requirements. We hope that the journal will be able to correct this when they undertake the final copy-editing process.

The title for Table 2 also appears to have been lost in formatting changes. We have attempted to add the title of the table back in but cannot do so without altering the formatting the journal has inserted. We’ve added in the following title at the bottom of the table and would ask that the journal please amend as necessary: ‘Table 2: Study location and participant demographic data’.

Line 136  Envenomation, had to look it up.

No response

Lines 137-218    Data on snake bites very interesting.

Thank you for your feedback.

Lines 202-17 treatment snake bites

No response

Lines 219 - 259 heavy metals.  Like comment on Se being a non-metal (below Line 238), it is a metalloid.

Thank you for your feedback.

260-347 Drinking and drugs  Excellent section which most all countries have data on.

Thank you for your feedback.

Line 328   however “quality data on levels of alcohol use or risk of alcohol poisoning are also lacking.”  Any Ideas?

Unfortunately, the Myanmar government has not undertaken a systematic assessment of drug or alcohol use across the country. Until the Government undertakes such research, the level of reliable data will remain low. However, given recent political developments in Myanmar and strikes on the part of healthcare/government staff, such research is unlikely to be done in the current context.

Lines 348-368  Agrochemicals  Short section, but has one emphasis on organo phosphate fertilizers yielding low sperm count.

No response required

Lines 369 -  382  Traditional Medicine  short  One case study of Dan Ywet.

No response required

Lines 383 – 474 Discussion  Identifies that the manuscript is the “first systematic assessment of the literature pertaining to poisoning in Myanmar.”  Included is data on health issues from Vietnam, Thailand.  This section discusses the results of a recent thesis which focuses on the reports from the New Yangon General Hospital, Poison Treatment Unit’s records.  The Unit does not generally treat snake bites, so data missing on this topic.  Most common treatment is for drug overdose.

Line 446 (there is s 2 in the word overdose2).

Thank you for your comment. The ‘2’ is an in-text citation for a footnote. The current formatting has been an amendment made by the journal from our initial submission, so we’ll refrain from changing it.

L   475-489 Conclusions-  Looks to me that the conclusions point to the is a need for more quality research with peer reviewed reporting.

Thank you, you are correct, and we hope that this review will help researchers in Myanmar identify areas of need.

Reviewer 4 Report

Thanks for your wonderful paper my only concern is a lack of explanation of the health effects of the exposures of interest in either the introduction of discussion.

For example,

  • Snakebites

Adukauskienė, Dalia, Eglė Varanauskienė, and Agnė Adukauskaitė. "Venomous snakebites." Medicina 47, no. 8 (2011): 461.

Hifumi, T., Sakai, A., Kondo, Y., Yamamoto, A., Morine, N., Ato, M., Shibayama, K., Umezawa, K., Kiriu, N., Kato, H. and Koido, Y., 2015. Venomous snake bites: clinical diagnosis and treatment. Journal of intensive care, 3(1), pp.1-9.

Del Brutto, O.H. and Del Brutto, V.J., 2012. Neurological complications of venomous snake bites: a review. Acta Neurologica Scandinavica, 125(6), pp.363-372.

  • Heavy metals

Obeng-Gyasi, E., 2020. Chronic cadmium exposure and cardiovascular disease in adults. Journal of Environmental Science and Health, Part A, 55(6), pp.726-729.

Navas-Acien, A., Guallar, E., Silbergeld, E.K. and Rothenberg, S.J., 2007. Lead exposure and cardiovascular disease—a systematic review. Environmental health perspectives, 115(3), pp.472-482.

Zheng, L., Kuo, C.C., Fadrowski, J., Agnew, J., Weaver, V.M. and Navas-Acien, A., 2014. Arsenic and chronic kidney disease: a systematic review. Current environmental health reports, 1(3), pp.192-207.

Drugs of abuse

Leshner, A.I., 1997. Addiction is a brain disease, and it matters. Science, 278(5335), pp.45-47.

Vasica, G. and Tennant, C.C., 2002. Cocaine use and cardiovascular complications. Medical Journal of Australia, 177(5), pp.260-262.

 agro-chemicals

Sekhotha, M.M., Monyeki, K.D. and Sibuyi, M.E., 2016. Exposure to agrochemicals and cardiovascular disease: a review. International journal of environmental research and public health, 13(2), p.229.

and traditional medicine.

Liu, J., Shi, J.Z., Yu, L.M., Goyer, R.A. and Waalkes, M.P., 2008. Mercury in traditional medicines: is cinnabar toxicologically similar to common mercurials?. Experimental biology and medicine, 233(7), pp.810-817.

Author Response

Reviewer four:

'Thanks for your wonderful paper my only concern is a lack of explanation of the health effects of the exposures of interest in either the introduction of discussion.'

Response: Thank you very much for your kind comments. We appreciate your feedback on the manuscript and have made the following changes:

  • We’ve cited the paper below and made a reference to neurological symptoms associated with venomous snake bite. The sentence reads: ‘Snake venom toxins are neurotoxic and interfere with blood coagulation, which can result in symptoms ranging from prolonged bleeding to stroke and neurological damage [51]’. We hope that other outcomes of snakebite have been adequately covered while discussing symptoms and patient outcomes.

Del Brutto, O.H. and Del Brutto, V.J., 2012. Neurological complications of venomous snake bites: a review. Acta Neurologica Scandinavica125(6), pp.363-372.

  • We’ve added the following sentence to introduce the section on heavy metals: ‘Environmental exposure to heavy metals has been attributed to a number of deleterious health conditions including but not limited to kidney disease, cardiovascular disease, neurological impairment and poor development in children [52-54]’. We’ve cited two of the articles you’ve provided, as well as a third article that discusses developmental and neurotoxic effects:

Obeng-Gyasi, E., 2020. Chronic cadmium exposure and cardiovascular disease in adults. Journal of Environmental Science and Health, Part A55(6), pp.726-729.

Zheng, L., Kuo, C.C., Fadrowski, J., Agnew, J., Weaver, V.M. and Navas-Acien, A., 2014. Arsenic and chronic kidney disease: a systematic review. Current environmental health reports1(3), pp.192-207.

Wasserman, G.A., Liu, X., Factor‐Litvak, P., Gardner, J.M. and Graziano, J.H. (2008), Developmental Impacts of Heavy Metals and Undernutrition. Basic & Clinical Pharmacology & Toxicology, 102: 212-217. https://doi.org/10.1111/j.1742-7843.2007.00187.x

  • We’ve added the following sentence to introduce the section on pesticides: ‘Exposure to pesticides can lead to acute or chronic poisoning, with outcomes ranging from irritation of the skin and the eyes to impaired development in children, reproductive disorders, cancer and death [65]’. We’ve used the reference below as it discusses the fact that pesticide poisoning experienced at higher levels in developing countries.

Kesavachandran, C. N., Fareed, M., Pathak, M. K., Bihari, V., Mathur, N., & Srivastava, A. K. (2009). Adverse health effects of pesticides in agrarian populations of developing countries. Reviews of environmental contamination and toxicology Vol 200, 33-52.

  • Thank you very much for suggesting a reference that discusses mercury contamination for the section on traditional medicine. However, we hope the detrimental health effects of heavy metals is now adequately highlighted in the accompanying heavy metals section.

Reviewer 5 Report

The authors of this research performed a systematic review about poisoning and toxic exposures in Myanma. The authors included 34 papers in this systematic review. It does not present innovation, it does not add scientific information that justifies its publication. It sometimes gives an almost childlike description of how the intoxication happened. This review adds nothing new to what can be done to improve the problem of poisoning, its recording, its monitoring.

Minor comment:

The reference list is not cited in accordance with the journal's proposal.

Author Response

We thank the reviewer for taking the time to read the manuscript. Unfortunately, the reviewer does not substantiate the negative report with any evidence or examples from the manuscript. The review is neither academically sound nor fair. We, therefore, have decided not to respond specifically and request that the reviewer either writes a proper review supported by examples and evidence from the manuscript or this review is discounted by the Editor. Reviewers 1 to 4 were highly supportive of the manuscript and the manuscript has been revised according to their suggestions. 

Round 2

Reviewer 1 Report

The authors have addressed my concerns satisfactorily. I recommend this paper for further processing in the journal.

Reviewer 4 Report

Edits are acceptable.